# Impact of COVID-19 Pandemic on Health-Related Quality of Life and Physical Activity of Patients in Hemodialysis

**DOI:** 10.3390/healthcare10112217

**Published:** 2022-11-04

**Authors:** Eva Segura-Ortí, Francisco J. Martínez-Olmos, Álvaro Ródenas-Pascual, Edgar Guillem-Giménez, Verónica Vercher-Narbona, Martha Jocelyne Piñón-Ruiz, Alicia Garcia-Testal

**Affiliations:** 1Department of Physiotherapy, Universidad Cardenal Herrera-CEU, CEU Universities, 46001 Valencia, Spain; 2Department of Nephrology, Hospital de Manises, 46001 Valencia, Spain

**Keywords:** health-related quality of life, physical activity, hemodialysis, COVID-19, social adjustment

## Abstract

Chronic dialysis patients have an increased risk of severe COVID-19 infection-related complications. The aim of this study was to quantify the impact of the COVID-19 pandemic on health-related quality of life (HRQoL) and physical activity levels of patients undertaking hemodialysis (HD). This was an observational study that compared data from two periods of time, before the COVID-19 pandemic vs pandemic. We used the Medical Outcomes Survey Short Form (SF-36) to measure the HRQoL and the Human Activity Profile (HAP) questionnaire was used to measure the physical activity. Data were analyzed with a mixed ordinal linear regression. A total of 27 eligible participants were interviewed during COVID-19 pandemic (median age 78 years). The linear regression model showed that the pandemic, after controlling for the covariates age, comorbidity, albumin, and hemoglobin, had a significant impact on the HRQoL. Physical function (−15.7) and social functioning subscales (−28.0) worsened (*p* = 0.001), and the physical component scale also showed a significant decrease (−3.6; *p* = 0.05). Time had a significant impact on the Human Activity Profile, with an average activity score diminished with the pandemic (−13.9; *p* = 0.003). The COVID-19 pandemic had a very negative impact on HRQoL and physical activity level of subjects undertaking hemodialysis. Interventions to improve HRQoL and activity levels of patients undertaking HD are recommended.

## 1. Introduction

The World Health Organization (WHO) declared a global pandemic on March 2020, in response to the sudden outbreak the COVID-19, which spread quickly from China across the world [1] and had a very negative impact on the medical activity [2]. Experience from previous pandemics has shown that exposure to the SARS virus results in feeling fear and guilt about exposing families to infection loneliness, anxiety, and depression [3]. Reasons include family separation during national lockdowns to contain virus spread and fear, and vulnerability [3]. Older chronic dialysis patients, particularly those aged 60–79 years old, had an increased risk of severe COVID-19 infection-related complications [4]. The psychological pressure has resulted in negative psychological emotions such as anxiety and depression mainly in in-center maintenance hemodialysis [1], due to increased COVID-19 infection exposure during travel to the dialysis centers, and having lack of control over dialysis treatment [5].

Health-related quality of life (HRQoL) is a multidimensional concept that includes several domains focused on physical, mental, emotional, and social functioning and reflects the impact of health status on individual well-being [6]. One previous study showed that chronic kidney disease (CKD) affects patients’ HRQoL, so subjects in the more advanced stages of CKD have worse HRQoL than those at the earlier stages of the disease [7]. Low HRQoL is associated with clinical outcomes, increased chance of hospitalization, and decreased survival in subjects undertaking dialysis [8]. HRQoL was negatively impacted by the pandemic in the general population [9], but also in people suffering from CKD [10].

Physical activity (PA) is any body movement produced by muscles that result in increased energy expenditure [5]. The PA level is lower in chronic kidney disease (CKD) patients at any stage compared with healthy counterparts [11]. Intradialysis exercise has proven to be beneficial for subjects undertaking hemodialysis. Conventional exercise includes aerobic (cycling) and strength training of the lower limb muscles [12,13].

During confinement, citizens were requested to practice social distancing (maintaining 2 m between themselves and other people) and avoid social gatherings. With these public measures, it would be expected that the population would be faced with changes in their physical activity behavior [14], and several authors suggest that the COVID-19 pandemic may lead to the worsening of the pandemic of inactivity, supported by the closure of recreation facilities, city parks, and playgrounds, and requiring innovative strategies to keep the physical activity practices [15]. A recent meta-analysis concluded that physical activity declined during COVID-19 pandemic lockdown regardless of the subpopulation (healthy or participants who had medical conditions). Guidelines recommend a minimum of 150 min per week of physical activity at a moderate intensity [16]. Subjects undertaking hemodialysis (HD) treatment are sedentary, they may spend almost 18 h/day in a sitting or reclining posture [17]. Sedentary behavior (relative time that subjects were in a sitting or reclining posture according to accelerometer data) was an important factor for all-cause mortality and was negatively associated with HRQoL. There was a clinical impact of sedentary behavior on HRQoL even after adjusting for confounding factors such as sex, duration of hemodialysis, age, and history of diabetes mellitus [17]. 

The aim of this study was to quantify the impact of the COVID-19 pandemic on HRQoL and the physical activity levels of subjects undertaking hemodialysis. We hypothesize that COVID-19 would negatively impact HRQoL and the physical activity level.

## 2. Materials and Methods

This was an observational study that compared data from two periods of time, before the COVID-19 pandemic (November 2019 to February 2020) vs pandemic (November 2020 to February 2021), regarding HRQoL and physical activity level.

The present study includes patients from the REalidad Virtual IntraDyalisis (REVID) study. REVID is a currently ongoing randomized trial that investigates the impact of intradialysis exercise. Participants in REVID are enrolled from September 2019 onwards. Thus, the data were recorded before the pandemic, and no more data were recorded until November 2020 once the situation at the dialysis units was under control.

Participants were recruited from the HD unit (source deleted for blinded review). Patients receiving HD treatment for more than 3 months and with a stable medical condition were included. The exclusion criteria were myocardial infarction (in the 6 weeks prior), amputation of a lower limb below the knee, cerebrovascular disease, patients suffering from chronic cardiac or respiratory diseases that presented chest pain, dyspnea or others after exertion, and inability to perform the functional tests. 

All the participants gave their written informed consent to participate in the study, which was approved by the Ethics Committee (source deleted for blinded review). (registration number 2018/0633). This study was registered at www.clinicaltrials.gov (NCT04046042). The RCT study started on September 2019 and stopped in March 2020 due to the pandemic. HRQoL and physical activity level data were registered pre-pandemic (from November 2019 to February 2020), and during pandemic (from November 2020 to February 2021).

### 2.1. Demographic, Anthropometric and Laboratory Data

Baseline demographic and clinical characteristics, including age, sex, body mass index, time on HD, and Charlson index were obtained pre-pandemic by review of medical records. The hemodialysis session details and the biochemical variables across the study were registered both on the pre-pandemic, and on the pandemic, including albumin, hemoglobin, creatinine and erythropoietin, among others. 

### 2.2. Health-Related Quality of Life and Physical Activity Level

Measures of health-related quality of life (HRQoL) and physical activity through questionnaires were registered twice, pre-pandemic while the ongoing RCT testing the impact of an intradialysis exercise program was implemented (from November 2019 to February 2020), and during pandemic (from November 2020 to February 2021) without any ongoing exercise program. All the participants were interviewed by a researcher or by a nurse during the HD sessions. 

HRQoL was assessed using the Medical Outcomes Study–Short Form 36-item (SF-36) questionnaire, Spanish version [18,19]. Scores range from 0 to 100, with higher scores reflecting better HRQoL. Results on 8 subscales (physical function, vitality, role physical, general health, social function, mental health, role mental, body pain) and 2 components (physical and mental component scale), were calculated.

Physical activity level was assessed using the Human Activity Profile (HAP) questionnaire which has been validated in populations with CKD [20]. This study showed that the results of HAP were correlated with accelerometer data, which is the gold standard for direct assessment of physical activity. Recent research shows that HAP discriminates among patients with higher and lower physical function [21]. The HAP comprises a list of 94 activities each assigned a number relating to the vigorousness of the task and the participants must report whether they still perform this activity, no longer perform the activity, or never performed the activity. The HAP assesses the maximal activity score (MAS, the number assigned to the activity with the highest oxygen consumption requirement that the patient still performs) and the adjusted activity score (ASS, the difference between the MAS and the number of activities with lower scores—i.e., less demanding tasks—that the participant has stopped performing). Depending on the AAS, individuals can be classified as having impaired activity (AAS < 53), being moderately active (AAS = 53–74), or as active (AAS > 74) [22]. The AAS has a high test–retest reliability in this population (ICC ASS = 0.92) [23]. HAP correlates with the result of physical function tests, such as the 6 MWT, STS10, and the 4-m gait speed test [21].

### 2.3. Statistical Analysis

Continuous variables are described as the mean (standard deviation) and median (1st and 3rd Quartile) and the qualitative variables are expressed as percentages.

To determine the extent to which the pandemic (time effect) was associated with the HRQoL (SF-36) or the physical activity score (AAS average activity score measured with the HAP), we used a mixed ordinal linear regression model. This model assessed if changes occurred in HRQoL and physical activity level between the two observation periods, and the covariables age, comorbidity (Charlson Index) hemoglobin, and albumin were included to control their effect on the dependent variables. To correct the non-independence of observations, participants were included as a random factor, so that the baseline level of each subject was considered in the analysis. A scatterplot was used to assess if there was a relationship between the variables before including them into the model. These statistical analyses were carried out with R-COMANDER considering probability values of *p* < 0.05 as significant. 

## 3. Results

Out of the 36 participants in the REVID study, 30 eligible participants were interviewed during COVID-19 pandemic. Figure 1 shows the flow diagram of participants, with a final sample of 27 participants (21 respondents for HAP, 21 respondents for the SF-36).

Table 1 shows the baseline demographic and clinical characteristics of the sample, with more men than women (63%), a median age of 78 years, a vintage in HD treatment of a median of 4 years and a high comorbidity index (median 8). 

Table 2 shows the details of the HD session in the two observational periods, pre-pandemic and pandemic and the biochemical characteristics across the study. Hemoglobin and albumin were posteriorly included in the model to explain changes in the dependent variables as covariates. 

Table 3 describes the pre-pandemic and post-pandemic data of SF-36 and HAP questionnaires. 

The lineal regression model (Table 4) showed that the pandemic (time), after controlling for the covariates age, comorbidity (Charlson Index), albumin (as an indicative of nutritional status), and hemoglobin, had a significant impact on the HRQoL measured through the SF-36. Time had an impact on two of the variables analyzed, the physical function subscale, and the social function subscale, but also for the physical component scale (PCS) measured with the SF-36 almost reached significance (*p* = 0.05). Time did not show a significant impact on the rest of the scales or components of the SF-36. The same model showed that time had a significant impact on the average activity score (AAS) measured through the Human Activity Profile.

## 4. Discussion

This study quantifies the negative impact of the COVID-19 pandemic on several dimensions of HRQoL and the physical activity level of subjects undertaking hemodialysis. To our knowledge, this is the first study that shows how negatively the pandemic has impacted this cohort regarding the different subscales of HRQoL measured through the SF-36 and to what extent the physical activity level is decreased. In addition, this is the only study that compares scores of these variables before and after the pandemic.

All subscales of SF-36 worsened during the pandemic, except for vitality. Two subscales (physical function and social function) and the PCS (*p* = 0.05) showed a significant decrease. It may be assumed that the negative impact was seen also in all populations, but some data show the opposite. Lockdown had a positive impact on HRQoL in young healthy soccer players, which showed a significant improvement in general and mental health [24]. A recent study showed that SF-36 scores, except for mental health in CKD patients, were significantly lower than healthy individuals during the pandemic [25].

Physical function decreased by more than 15 points and the PCS decreased by over 3.6 points during the observation period, comparing pre-pandemic scores and scores during the pandemic. The negative impact of COVID-19 on HRQoL has been shown in CKD patients (stages 4 and 5) aged 70 years or older before dialysis initiation [10]. Similarly, our study also found no changes during the COVID-19 pandemic on the mental component scale, but there was a decrease in PCS that was not associated with patient characteristics except for sex (males showed a greater decline −5.3 SD 8.5 compared to females −0.9 SD 5.7). The PCS and the physical function subscale of the SF-36 are commonly reported when interventions are focused on the functional capacity of people undertaking HD. Literature reports the results of exercise interventions aimed at improving both of them [6,26]. Thus, the significant negative impact of the pandemic on the PCS and the physical function subscale is a very negative objectivation of how harmful the pandemic has been. Both the PCS and the physical function subscale correlate with test results such as the 6 MWT [27]. These results suggest that interventions to revert the PCS and the physical function subscale to pre-pandemic levels, or even to improve them are of great need.

Social activities during the pandemic decreased; during confinement, it was not possible to stay out of home, and only visits to the hospital for HD treatment were allowed. Once the confinement finished, social activities were limited due to restrictions on social interaction that varied accordingly to the incidence rates. However, interactions’ limitation was also self-imposed before and after vaccination occurred due to fear of the virus contamination. Thus, it seems sensible that social impact was severely affected by the pandemic. Lockdown duration and, therefore, lack of social interaction increases psychological distress in the general population [28]. The pandemic increased the prevalence of fatigue, anxiety, depression, and poor sleep quality among subjects undertaking hemodialysis [1,29].

The literature shows a negative impact of the pandemic on several psychological aspects, including decreased HRQoL in subjects with diabetes mellitus [30] and chronic neurological diseases [31], but also in enterprises workers [9] or pregnant women [32] and even poor sleep in the general population [33].

On the other side, an expected negative impact of the pandemic on physical activity level has been shown, since the average activity score of the HAP showed a significant decrease. This questionnaire is a valid tool to measure physical activity [20] and it correlates with the functional capacity of patients undertaking HD [21]. It was an expected impact of the pandemic that has been shown in the current sample. This finding agrees with the results of a meta-analysis that concluded that both healthy subjects and subjects suffering from medical conditions showed a decreased physical activity level after the pandemic. Stockwell et al. recommend the promotion of digital-based PA [34]. The mobility restrictions imposed by the pandemic situation added to the self-imposed restrictions due to fear of contracting the virus, resulting in a decreased daily activity of the general population. Thus, some authors talk about the silent pandemic of sedentarism [15] and its consequences are as scary as the virus itself. Different authors report the physical and psychological impact of this immobility in other cohorts.

The changes seen in the patients’ PA and HRQoL could be due to factors that are already known that contribute to/are associated with low activity levels and poor HRQoL among patients with CKD receiving HD. Nevertheless, previous research of our group showed that 6 months did not negatively impact in a significant decrease in the physical function of this cohort [35].

There are several limitations of this study. This is an observational study that is not adding data on prevalence/incidence of COVID-19 among the study population, though despite suffering or not from COVID-19, participants, as well as the rest of the population in the world, were affected by the fact of the pandemic itself. A selection bias exists in patients that volunteered to participate in an exercise program, these participants may have a higher functional capacity and/or psychological state that translate into active participation in in-center exercise programs. Thus, the representativeness of the sample could be compromised. There is a limited sample size and for this reason, different variables such as type of access, number of dialysis sessions per week, blood flow, and the dialysis technique could not be considered in the analysis. Participants were included in a study that was terminated by COVID-19, and this fact could have had an impact on results, given that the intervention may have helped develop habits and motivation for participants to continue exercising on their own. Responses were given during the dialysis treatment, trying to achieve as much anonymity and privacy as possible, but it was still limited and this fact could have influenced responses. There is a great variety regarding how each country responded to the pandemic crisis during the studied time period regulatory policies and social/ cultural reactions. Finally, the time frame of testing before and after the pandemic is quite wide due to the unexpected nature of the pandemic and the clinical difficulties of managing the pandemic situation in in-center hemodialysis. Although it was not possible to administer questionnaires among the recruited HD patients between March 2020 and November 2020, other recent findings suggest that these patients may be negatively affected by the pandemic due to largely possible unmet care needs [36]. Therefore, the results should be coherent without the missing time period.

## 5. Conclusions

In conclusion, it seems that the COVID-19 pandemic had a very negative impact on HRQoL and on the physical activity level of subjects undertaking hemodialysis. Future studies should show if vaccination and the step-by-step back to normal life have had an impact on reversing this situation. Interventions to improve HRQoL and activity levels of patients undertaking HD is now an emergency.

## Figures and Tables

**Figure 1 healthcare-10-02217-f001:**
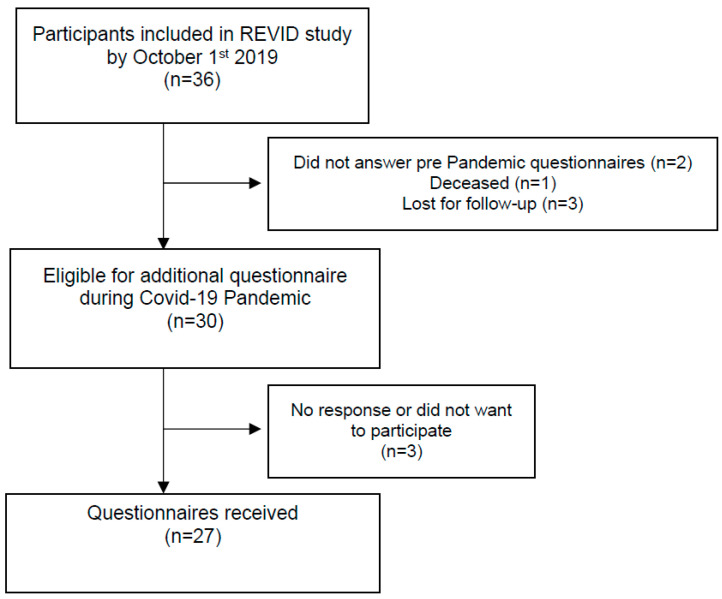
Flow Diagram.

**Table 1 healthcare-10-02217-t001:** Baseline clinical characteristics and demographics.

Variable (n = 27)	Demographics	
Age (years)		
Mean (SD)	73.4	(14.1)
Median (1st Q–3rd Q)	78	(68.5–82)
Sex n (%)		
Male	17	(63.0%)
Female	10	(37.0%)
Body Mass Index (kg/m^2^)		
Mean (SD)	27.8	(6.5)
Median (1st Q–3rd Q)	25.3	(23.7–30.3)
Time on HD (years)		
Mean (SD)	7.0	(6.0)
Median (1st Q–3rd Q)	4	(3–9)
Charlson’s Comorbidity		
Mean (SD)	7.8	(2.4)
Median (1st Q–3rd Q)	8	(6–9)
Diabetes n (%)		
No	15	(44.4%)
Yes	12	(55.6%)
Hypertension n (%)		
No	2	(7.4%)
Yes	25	(92.6%)
Cardiopathy n (%)		
No	8	(29.6%)
Yes	19	(70.4%)
COPD n (%)		
No	23	(85.2%)
Yes	4	(14.8%)
Neoplasia n (%)		
No	21 (77.8%)	21 (77.8%)
Yes	6 (22.2%)	6 (22.2%)
Blood flow (mL/min)		
Mean (SD)	364.2	(36.1)
Median (1st Q–3rd Q)	350	(344–394)
Number of HD (sessions/week)		
2	2	(7.4%)
3	25	(92.6%)
Vascular access		
Arteriovenous fistula	22	(81.5%)
Central Venous Catheter	5	(18.5)
Dialysis technique		
HDHF	17	(63%)
HFOL	9	(33.3%)
HDLF	1	(3.7%)

Legend: COPD, Chronic obstructive pulmonary disease; HD, Hemodialysis; HDHF, High-flow hemodialysis; HDLF, Low flow hemodialysis; HFOL, Online hemodiafiltration; Q, Quartile; SD, Standard Deviation.

**Table 2 healthcare-10-02217-t002:** Hemodialysis session details and biochemical characteristics across the study.

**Hemodialysis Session**	**Pre-Pandemic**		**Post-Pandemic**	
Session duration (minutes)				
Mean (SD)	230.2	(8.6)	231.2	(7.2)
Median (1st Q–3rd Q)	232	(230–233)	232	(231–234)
Dry weight (Kg)				
Mean (SD)	70.3	(14.6)	67.6	(15.2)
Median (1st Q–3rd Q)	68.5	(59.7–77.1)	64.5	(57.4–74.5)
KtV sp				
Mean (SD)	1.49	(0.23)	1.61	(0.25)
Median (1st Q–3rd Q)	1.50	(1.31–1.66)	1.55	(1.42–1.75)
Heart rate (beats per minute)				
Mean (SD)	67	(9)	68	(9)
Median (1st Q–3rd Q)	68	(61–75)	68	(63.5–74.5)
Systolic blood pressure (mmHg)				
Mean (SD)	143	(16)	145	(22)
Median (1st Q–3rd Q)	141	(135–154)	147	(130–162)
Diastolic blood pressure (mmHg)				
Mean (SD)	60	(10)	60	(11)
Median (1st Q–3rd Q)	59	(52–67)	59	(53–67)
**Biochemical Characteristics**	**Pre-Pandemic**		**Post-Pandemic**	
Hemoglobin (g/dL)				
Mean (SD)	11.3	(0.9)	11.6	(1.2)
Median (1st Q–3rd Q)	11.2	(10.8–11.6)	11.8	(11.2–12.5)
Albumin (g/dL)				
Mean (SD)	4.1	(0.3)	3.9	(0.4)
Median (min–max)	4.0	(3.8–4.2)	3.9	(3.8–4.2)
Phosphorus (mg/dL)				
Mean (SD)	4.3	(1.7)	31.3	(18.6)
Median (1st Q–3rd Q)	4.1	(3.2–5.4)	37	(10–48)
Ferritin (ng/mL)				
Mean (SD)	525.6	(312.0)	539.5	(291.2)
Median (1st Q–3rd Q)	482	(313–729.5)	613	(297.5–760)
PTH intacta (pg/mL)				
Mean (SD)	273.8	(200.9)	244.2	(126.3)
Median (1st Q–3rd Q)	266	(163.5–308.5)	218	(178–318)
Calcium (mg/dL)				
Mean (SD)	9.1	(0.8)	8.5	(0.6)
Median (1st Q–3rd Q)	9.1	(8.5–9.4)	8.3	(8.1–8.9)
Phosphorus (mg/dL)				
Mean (SD)	4.3	(1.7)	31.3	(18.6)
Median (1st Q–3rd Q)	4.1	(3.2–5.4)	37	(10–48)

Legend: Post-Pandemic, November 2020 to February 2021; Pre-Pandemic, November 2019 to February 2020; Q, Quartile; SD, Standard Deviation.

**Table 3 healthcare-10-02217-t003:** Health-related quality of life and physical activity level of hemodialysis patients.

**Health-Related Quality of Life**	**Pre-Pandemic**		**Post-Pandemic**	
SF-36 Physical Function				
Mean (SD)	52.1	(30.1)	36.4	(34.3)
Median (1st Q–3rd Q)	50	(30–80)	30	(5–75)
SF-36 Role Physical				
Mean (SD)	52.4	(46.0)	39.3	(43.7)
Median (1st Q–3rd Q)	75	(0–100)	25	(0–75)
SF-36 Bodily Pain				
Mean (SD)	64.8	(40.8)	61.4	(33.3)
Median (1st Q–3rd Q)	100	(31–100)	62	(41–100)
SF-36 General Health				
Mean (SD)	46.6	(17.5)	38.9	(22.8)
Median (1st Q–3rd Q)	45	(40–57)	30	(30–50)
SF-36 Vitality				
Mean (SD)	46.7	(29.7)	50.2	(25.1)
Median (1st Q–3rd Q)	50	(20–65)	50	(40–70)
SF-36 Social Functioning				
Mean (SD)	76.2	(33.3)	48.2	(36.3)
Median (1st Q–3rd Q)	100	(62.5–100)	37.5	(25–87.5)
SF-36 Role Emotional				
Mean (SD)	76.2	(41.0)	57.1	(42.4)
Median (1st Q–3rd Q)	100	(66.7–100)	66.7	(0–100)
SF-36 Mental Health				
Mean (SD)	71.0	(27.0)	64.8	(22.7)
Median (1st Q–3rd Q)	80	(48–92)	64	(52–84)
SF-36 Physical Component Scale				
Mean (SD)	37.5	(11.1)	34.0	(11.6)
Median (1st Q–3rd Q)	35.5	(29.3–48.2)	29.9	(26.3–41.7)
SF-36 Mental Component Scale				
Mean (SD)	49.1	(13.3)	43.8	(12.3)
Median (1st Q–3rd Q)	52.6	(40.1–59.5)	39.6	(34.5–54.8)
**Activity Level**	**Pre-Pandemic**		**Post-Pandemic**	
Average Activity Score				
Mean (SD)	46.4	(26.9)	32.5	(27.4)
Median (1st Q–3rd Q)	51	(28–71)	28	(7–52)
Maximal Activity Score				
Mean (SD)	64.0	(22.5)	45.3	(30.1)
Median (1st Q–3rd Q)	69	(67–75)	63	(12–71)

Legend: Post-Pandemic, November 2020 to February 2021; Pre-Pandemic, November 2019 to February 2020; Q, Quartile; SD, Standard Deviation; SF-36, Medical Outcomes Survey Short Form.

**Table 4 healthcare-10-02217-t004:** Linear regression model.

	Estimate (CI 95%)	Standard Error	*p* Value
SF-36 Physical Function	−15.714 (−24.245, −7.183)	4.258	0.001
SF-36 Role Physical	−13.095 (−37.911, 11.721)	12.514	0.308
SF-36 Bodily Pain	−3.429 (−14.86, 8.003)	5.705	0.555
SF-36 General Health	−7.762 (−15.898, 0.374)	4.061	0.07
SF-36 Vitality	3.571 (−8.458, 1.812)	4.501	0.437
SF-36 Social Functioning	−27.976 (−42.631, −13.321)	7.314	0.001
SF-36 Role Emotional	−19.047 (−41.42, 3.325)	11.168	0.104
SF-36 Mental Health	−6.238 (−15.082, 2.606)	4.414	0.173
SF-36 Physical Component Scale	−3.569 (−7.083, −0.055)	1.754	0.05
SF-36 Mental Component Scale	−5.37 (−10.816, 0.076)	2.718	0.062
HAP Average Activity Score	−13.905 (−22.282, −5.528)	4.181	0.003

Legend: CI, Confidence interval; HAP, Human Activity Profile; Post-Pandemic, November 2020 to February 2021; Pre-Pandemic, November 2019 to February 2020; Q, Quartile; SD, Standard Deviation; SF-36, Medical Outcomes Survey Short Form.

## Data Availability

The original contributions presented in the study are included in the article; further inquiries can be directed to the corresponding author.

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
