# Peer review of "Impact of COVID-19 Pandemic on Health-Related Quality of Life and Physical Activity of Patients in Hemodialysis"

_healthcare, 2022, doi:10.3390/healthcare10112217_

Round 1

Reviewer 1 Report

Dear Author,

It is delightful to have the opportunity to review "Impact of COVID-19 Pandemic on Health-Related Quality of 2 Life and Physical Activity of Patients in Hemodialysis."

I hope my opinions will contribute to the publication of the high-quality paper:

1) The sample was n=27. I believe it is more honest to refer in the abstract to the number of the sample and not the number of eligible participants.

2) Lines 31-32 "Global pandemics have a significant effect on mental well-being, including fear of contraction and spreading the infection to family members, loneliness, anxiety, and depression." Add the references of the studies that support this sentence.

3) Line 43 – "Several studies..."

Although there is only one reference at the end of the sentence ([6] Nguyen, N.T.Q.; Cockwell, P.; Maxwell, A.P.; Griffin, M.; O'Brien, T.; O'Neill, C. Chronic kidney disease, health-related quality 292 of life and their associated economic burden among a nationally representative sample of community dwelling adults in 293 England. PLoS One 2018, 13, e0207960, 10.1371/journal.pone.0207960 [doi].)

4) Line 47 – the same comment. “several studies,” although we just read one reference.

5) Lines 52-54 There is more recent literature about the subject, for example, PMID: 31454822

6) Lines 66 – Several authors, although present one reference.

7) The number of ethical commitments is from 2017, so I believe that it is important to clarify how you collect data to give answers for hypotheses during the pandemic period. Also, because this important information is different from the information in the Institutional Review Board Statement

8) Line 107 – “All the participants were interviewed by a researcher or a nurse during the HD sessions.” Was the interview made close to the other patients? How were anonymity and privacy ensured?

9) How was the risk of bias reduced considering that variables such as the type of access, the number of dialysis sessions, blood flow, and the dialysis technique determine the quality of life of patients undergoing hemodialysis?

9) Lines 172-173 “Time had an impact on three of the analyzed variables, the physical function subscale, the social function subscale, and the physical component scale (PCS).”

Although in table 4, the p-value for the Physical component scale is p= 0.05, you refer to “considering probability values ​​of p< 0.05 as significant.” It´s necessary to clarify.

1)      The number of the sample is small for the use of simple linear regression (Tabachnick, B. and L. Fidell, (2007). Using Multivariate Statistics, USA, Pearson International.)

To use this model, the sample must show a relationship between the variables that can be translated through a linear mathematical model. Considering the sample size, a scatterplot is essential to guarantee this assumption.

The statistical data the authors provide do not allow us to confirm whether there is an adjustment to the linear model.

11) To support the last affirmation in the conclusions “and the support inside the units of physiotherapists is now more necessary than ever” it is necessary `to include in the introduction and discussion studies that refer to the advantages of physiotherapists in dialysis clinics.

I suggest:

1) The possibility of adding the term Social Adjustment to keywords

2) Table 2 – the variable albumin is repeat.

3) Table 2 – standardize the hyphen between range of median. Example: "(3.8- 4.2); (53-67)"

4) Add title “5. Conclusion”.

5) The dry weight and albumin value decrease in the post-pandemic period (Table 2). Do you believe that results may be related to the decrease in physical activity level and consequent loss of muscle tissue?

Finally, congratulations on the work and the subject matter.

Best Regards,

Author Response

Thank you very much for your time in reviewing the manuscript and for your wise comments. I copied your answer here and in red I reply. I hope you are satisfied with the changes.

Dear Author,

It is delightful to have the opportunity to review "Impact of COVID-19 Pandemic on Health-Related Quality of 2 Life and Physical Activity of Patients in Hemodialysis."

I hope my opinions will contribute to the publication of the high-quality paper:

1) The sample was n=27. I believe it is more honest to refer in the abstract to the number of the sample and not the number of eligible participants.

Thank you for your comment. You are right, so we change the number in the abstract. Now you can read “27 eligible participants were interviewed during COVID-19 pandemic”

2) Lines 31-32 "Global pandemics have a significant effect on mental well-being, including fear of contraction and spreading the infection to family members, loneliness, anxiety, and depression." Add the references of the studies that support this sentence.

Thank you for your comment. The reference has been added in the middle of two sentences, it was from the same authors, but now it is clearer than before.

“Experience from previous pandemics has shown that exposure to the SARS virus results in feeling fear and guilt about exposing families to infection loneliness, anxiety, and depression [2]. Reasons include family separation during national lockdowns to contain virus spread and fear, and vulnerability [2].”

3) Line 43 – "Several studies..."

Although there is only one reference at the end of the sentence ([6] Nguyen, N.T.Q.; Cockwell, P.; Maxwell, A.P.; Griffin, M.; O'Brien, T.; O'Neill, C. Chronic kidney disease, health-related quality 292 of life and their associated economic burden among a nationally representative sample of community dwelling adults in 293 England. PLoS One 2018, 13, e0207960, 10.1371/journal.pone.0207960 [doi].)

Thank you for your comment, we have changed the text and now you can read One previous study showed that chronic kidney disease (CKD) affects patients’ HRQoL, so subjects at the more advanced stages of CKD have worst HRQoL than those at the earlier stages of the disease [6].

4) Line 47 – the same comment. “several studies,” although we just read one reference.

Thank you for your comment, we have changed the text and now you can read “HRQoL was negatively impacted by the pandemic in the general population [8], but also in people suffering from CKD [9].”

5) Lines 52-54 There is more recent literature about the subject, for example, PMID: 31454822

Thank you for your comment, we have added the new reference suggested

13   Huang, M.; Lv, A.; Wang, J.; Xu, N.; Ma, G.; Zhai, Z.; Zhang, B.; Gao, J.; Ni, C. Exercise Training and Outcomes in Hemodialysis Patients: Systematic Review and Meta-Analysis. Am. J. Nephrol. 2019, 50, 240-254

6) Lines 66 – Several authors, although present one reference.

Thank you for your comment. We have removed part of the sentence to be more accurate and now you can read Sedentary behavior (relative time that subjects were in a sitting or reclining posture according to accelerometer data) was an important factor for all-cause mortality and was negatively associated with HRQoL. There was a clinical impact of sedentary behavior on HRQoL even after adjusting for confounding factors such as sex, duration of hemodialysis, age, and history of diabetes mellitus [15]. “

7) The number of ethical commitments is from 2017, so I believe that it is important to clarify how you collect data to give answers for hypotheses during the pandemic period. Also, because this important information is different from the information in the Institutional Review Board Statement

Thank you for your comment, we have just realized we confused the registration number, the right one is 2018/0633 (we have changed in within the main text too). The approval was given by the end of 2018. As explained in the manuscript “The present study includes patients from the REalidad Virtual IntraDyalisis (REVID) study. REVID is a currently ongoing randomized trial that investigates the impact of intradialysis exercise. Participants in REVID are enrolled from September 2019 onwards” we started testing participants by the end of 2019, but then the pandemic stopped the trial activity, but the activity and the HRQoL questionnaires serve as to show the impacto of the pandemic for this fragile cohort.

8) Line 107 – “All the participants were interviewed by a researcher or a nurse during the HD sessions.” Was the interview made close to the other patients? How were anonymity and privacy ensured?

Thank you for your comment. Testing of subjects undertaking hemodialysis is very complicated since they spend many times at the clinical center and they do not want to add extra testing time at different times to the dialysis treatment. We are very experienced in testing and interviewing patients within the dialysis unit. Patients are separated ones from the others; many of them use earplugs to listen to the personal TV or radio, and sometimes we use separators if convenient. But this could in fact be a limitation, the limitation in privacy and anonymity. So we add this point to the discussion. Now you can read a new sentence “Responses were given during the dialysis treatment, trying to achieve as much anonymity and privacy as possible, but it was still limited and this fact could have influenced responses.”

9) How was the risk of bias reduced considering that variables such as the type of access, the number of dialysis sessions, blood flow, and the dialysis technique determine the quality of life of patients undergoing hemodialysis?

Thank you for your comment. Since the sample size was so limited the number of variables selected in the model were limited. There is no doubt all the variables you highlight could have an influence. So we add it as a limitation to the study in the discussion and now you can read “There is a limited sample size and for this reason, different variables such as type of access, number of dialysis sessions per week, blood flow, and the dialysis technique could not be considered in the analysis.”

9) Lines 172-173 “Time had an impact on three of the analyzed variables, the physical function subscale, the social function subscale, and the physical component scale (PCS).”

Although in table 4, the p-value for the Physical component scale is p= 0.05, you refer to “considering probability values ​​of p< 0.05 as significant.” It´s necessary to clarify.

Thank you for your comment. The statistics were performed by a statistician that recommended highlighting them as significant, even though it was not strictly < but = to 0.05. We have changed the table and we add this borderline result in the comments of results. Now you can read in the results section Time had an impact on two of the variables analyzed, the physical function subscale, and the social function subscale, but also for the physical component scale (PCS) measured with the SF-36 almost reached significance (p=0.05).”

1)      The number of the sample is small for the use of simple linear regression (Tabachnick, B. and L. Fidell, (2007). Using Multivariate Statistics, USA, Pearson International.)

To use this model, the sample must show a relationship between the variables that can be translated through a linear mathematical model. Considering the sample size, a scatterplot is essential to guarantee this assumption.

The statistical data the authors provide do not allow us to confirm whether there is an adjustment to the linear model.

Thank you for your comment. This analysis was performed by a professional statistician using R-COMANDER, which is not the most extended program in publications (researchers doing statistics but we are not statisticians we use SPSS, and SAS). For this piece of research, we trusted the statistician professional criteria totally, but it is true that sometimes in statistics different opinions exist. We are sorry this criterion is not in agreement with the reference you provide. Before the inclusion of the variables into the model there was a preliminary exploration of relationship between variables that included the scatterplot, but it was not reported by the professional statistician, maybe because it was assumed in his opinion. We add this information in the statistical analysis, now you can read “A scatterplot was used to assess if there was a relationship between the variables before including them into the model”.

11) To support the last affirmation in the conclusions “and the support inside the units of physiotherapists is now more necessary than ever” it is necessary `to include in the introduction and discussion studies that refer to the advantages of physiotherapists in dialysis clinics.

Thank you for your comment. You are right, and we cannot introduce this suggestion without adding any previous references. We consider the message is clear and the sentence can be removed, in order to avoid major changes in the extension of the manuscript. In conclusions now you can read Interventions to improve HRQoL and activity levels of patients undertaking HD is now an emergency”, and the support inside the units of physiotherapists is now more necessary than ever.

I suggest:

1) The possibility of adding the term Social Adjustment to keywords

Thank you, we added it

2) Table 2 – the variable albumin is repeat.

Thank you, we corrected it

3) Table 2 – standardize the hyphen between range of median. Example: "(3.8- 4.2); (53-67)"

Thank you, we standardized it

4) Add title “5. Conclusion”.

Thank you, we added it

5) The dry weight and albumin value decreased in the post-pandemic period (Table 2). Do you believe that results may be related to the decrease in physical activity level and consequent loss of muscle tissue?

Thank you for your comment. No, we do not believe these decreases were related to the physical activity level. You can read that albumin was included in the model as a covariable, so for sure it did not have an impact. We did not include dry weight, again the sample is too limited to add too many variables into the model.

Finally, congratulations on the work and the subject matter.

Thanks to you for your time and kindness in giving so much feedback after your extensive review. We hope you are satisfied with the changes and explanations that are given.

Best Regards,

Reviewer 2 Report

“This was an observational study that compared data from two periods of time, before the COVID-19 pandemic (November 2019 to February 2020) vs the pandemic (November 2020 to February 2021), regarding HRQoL and physical activity level.”

The authors should provide the rationale for excluding the time between March 2020 to November 2020. As the authors stated in the first sentence of the introduction, the global pandemic started in March 2020.

The main conclusion of this paper is that the pandemic negatively impacted HRQOL and physical functioning. But considering the fact that 1) this is an observational study, 2) March 2020-November 2020 was excluded, authors should be more circumspect about their findings. It may be more appropriate to interpret the results as the differences between the two time period. In particular, the fact that no information was provided on the prevalence/incidence of COVID-19 among the study population during this specific time period prevents this study from further investigating on “the impact of the COVID-19 pandemic”.

Also, the authors need some editing of the English language and style. For example, in the abstract. “The SF-36 was used to measure the HRQoL and the Human Activity Profile (HAP) questionnaire measured the physical activity.”

It should be something like “We used SF-36 to measure the HRQoL and the Human Activity Profile (HAP) questionnaire to measure the physical activity.” or “The SF-36 was used to measure the HRQoL and the Human Activity Profile (HAP) questionnaire was used to measure the physical activity.”

Also, the authors need to spell out SF-36 in the first use.

“Interventions to improve HRQoL and activity levels of patients undertaking HD are recommended.”

Need to spell out HD both in the Abstract and the main text. The readers can only guess that HD indicates Hemodialysis.

Author Response

Thank you very much for your time in reviewing the manuscript and for your comments. I copied your answer here and in red I reply. I hope you are satisfied with the changes.

“This was an observational study that compared data from two periods of time, before the COVID-19 pandemic (November 2019 to February 2020) vs the pandemic (November 2020 to February 2021), regarding HRQoL and physical activity level.”

The authors should provide the rationale for excluding the time between March 2020 to November 2020. As the authors stated in the first sentence of the introduction, the global pandemic started in March 2020.

Thank you for your comment. The methodology of the article includes the following information The present study includes patients from the REalidad Virtual IntraDyalisis (REVID) study. REVID is a currently ongoing randomized trial that investigates the impact of intradialysis exercise. Participants in REVID are enrolled from September 2019 onwards.” When the pandemic started we were in the middle of recruiting participants (data from November to February 2020). Once it started, the HD units were overwhelmed, so we did have no intention to register answers to questionnaires, some of the participants even died. When the situation at the HD units was a bit more stabilized, the nurses there started to register again the questionnaires (November 2020 to February 2021). We add some information to give rationale to the timing of the questionnaires, and now you can read “Thus, the data were recorded before the pandemic, and no more data were recorded until November 2020 once the situation at the dialysis units was under control.”

The main conclusion of this paper is that the pandemic negatively impacted HRQOL and physical functioning. But considering the fact that 1) this is an observational study, 2) March 2020-November 2020 was excluded, authors should be more circumspect about their findings. It may be more appropriate to interpret the results as the differences between the two time period. In particular, the fact that no information was provided on the prevalence/incidence of COVID-19 among the study population during this specific time period prevents this study from further investigating on “the impact of the COVID-19 pandemic”.

Thank you for your comment. You are right and we add some limitations regarding the observational nature of the study and the lack of data on incidence/prevalence. But here we do not pretend to explain how HRQoL and activity were affecting those patients that suffered COVID-19, but just the fact of living in a pandemic time. The whole population was affected by living such a period, but the question we want to answer is how specifically it affected this cohort, that in general presents lower levels of both HRQoL and activity than healthy counterparts. We really believe it is worthy of showing our results.

In the discussion, now you can read “This is an observational study that is not adding data on prevalence/incidence of COVID-19 among the study population, though despite suffering or not COVID-19 participants, as the rest of the population in the world, were affected by the fact of the pandemic itself.”

In the conclusion, we tried to change a bit the language and now you can read “In conclusion, it seems that the COVID-19 pandemic had a very negative impact on HRQoL and on the physical activity level of subjects undertaking hemodialysis”

Also, the authors need some editing of the English language and style. For example, in the abstract. “The SF-36 was used to measure the HRQoL and the Human Activity Profile (HAP) questionnaire measured the physical activity.”

It should be something like “We used SF-36 to measure the HRQoL and the Human Activity Profile (HAP) questionnaire to measure the physical activity.” or “The SF-36 was used to measure the HRQoL and the Human Activity Profile (HAP) questionnaire was used to measure the physical activity.”

Also, the authors need to spell out SF-36 in the first use.

Thank you very much for your comment. We have corrected the English style and we have spelled out SF-36 the first time we used in the abstract. Now you can read “We used the Medical Outcomes Survey Short Form (SF-36) to measure the HRQoL and the Human Activity Profile (HAP) questionnaire was used to measure the physical activity.

“Interventions to improve HRQoL and activity levels of patients undertaking HD are recommended.”

Need to spell out HD both in the Abstract and the main text. The readers can only guess that HD indicates Hemodialysis.

Thank you very much for your comment. We have spelled HD both in the Abstract and in the main text.

 Abstract: Chronic dialysis patients had an increased risk of severe COVID-19 infection-related complications. The aim of this study was to quantify the impact of the COVID-19 pandemic on health-related quality of life (HRQoL) and physical activity levels of patients undertaking hemodialysis (HD). This was an observational study that compared data from two periods of time, before the COVID-19 pandemic vs pandemic. We used the Medical Outcomes Survey Short Form (SF-36) to measure the HRQoL and the Human Activity Profile (HAP) questionnaire was used to measure the physical activity. Data were analyzed with a mixed ordinal linear regression. 27 eligible participants were interviewed during COVID-19 pandemic (median age 78 years). The linear regression model showed that the pandemic, after controlling for the covariates age, comorbidity, albumin, and hemoglobin, had a significant impact on the HRQoL. Physical function (-15.7) and social functioning subscales (-28.0) worsened (p=0.001), and the physical component scale also showed a significant decrease (-3.6; p=0.05). Time had a significant impact on the Human Activity Profile, with an average activity score diminished with the pandemic (-13.9; p=0.003). The COVID-19 pandemic had a very negative impact on HRQoL and on the physical activity level of subjects undertaking hemodialysis. Interventions to improve HRQoL and activity levels of patients undertaking HD are recommended

In the main text, introduction, you can read ·Subjects undertaking hemodialysis (HD) treatment are sedentary, they may spend almost 18h/day in a sitting or reclining posture [15].”

Reviewer 3 Report

Dear Editor,

I have read the paper by Safwan Omran et al with interest. The authors investigate the impact of the COVID-19 pandemic on HRQoL and the physical activity levels of subjects undertaking hemodialysis.

The Study found that the COVID-19 pandemic had a very negative impact on HRQoL and on the physical activity level of subjects undertaking hemodialysis. Interventions to improve HRQoL and activity levels of patients undertaking HD are recommended

The paper is of novelty and present interest for the community. However, before publication there are some concerns that must be resolved.

1. In the Introduction:

There is no need to explain where COVID-19 originates from; it is well-known everywhere around the world.

Also, please use „COVID-19” all over the manuscript, avoiding „coronavirus”.

Moreover, i suggest in the Introduction section to add more information regarding the negative impact of COVID-19 pandemic over the medical activity. See and add the following articles: https://doi.org/10.3390/jcm11092452 , https://doi.org/10.3389/fsurg.2022.883935

2. Please correct the following:

Line 21: impact on HRQOL and on the physical activity level -> impact on HRQOL and the physical activity level

Line 231:These authors recommend the promotion of digital-based PA”. Please write the authors with “XY et al. recommend”

… etc. Corrections need to be done all over the manuscript.

3. In the Discussion:

Lines 184-187: Avoid being repetitive. (To our knowledge, this is the first study that shows ……. . To our knowledge, this  is the only study )

4. Please add a Conclusion section, after the Discussions.

5. Revise the „Data Availability Statement”.

6. For References:

The references need to be in accordance with the instructions in the template. The list should be with 1., 2., 3., 4. … etc. Also, delete the “[doi]” from the reference list.

Please see the Authors’ Instructions: “In the text, reference numbers should be placed in square brackets [ ], and placed before the punctuation; for example [1], [1–3] or [1,3]. For embedded citations in the text with pagination, use both parentheses and brackets to indicate the reference number and page numbers; for example [5] (p. 10). or [6] (pp. 101–105). The reference list should include the full title, as recommended by the ACS style guide. Style files for Endnote and Zotero are available.”

Author Response

Thank you very much for your time in reviewing the manuscript and for your comments. I copied your answer here and in red I reply. I hope you are satisfied with the changes.

Dear Editor,

I have read the paper by Safwan Omran et al with interest. The authors investigate the impact of the COVID-19 pandemic on HRQoL and the physical activity levels of subjects undertaking hemodialysis.

The Study found that the COVID-19 pandemic had a very negative impact on HRQoL and on the physical activity level of subjects undertaking hemodialysis. Interventions to improve HRQoL and activity levels of patients undertaking HD are recommended

The paper is of novelty and present interest for the community. However, before publication there are some concerns that must be resolved.

  1. In the Introduction:

There is no need to explain where COVID-19 originates from; it is well-known everywhere around the world.

Thank you for your comments. We delete some sentences from the introduction according to your suggestions. Now you can read ”The World Health Organization (WHO) declared a global pandemic on March 2020, in response to the sudden outbreak the COVID-19, which spread quickly from China across the world [1].

Also, please use „COVID-19” all over the manuscript, avoiding „coronavirus”.

Thank you for your comments, we have removed ”coronavirus” from the manuscript

Moreover, i suggest in the Introduction section to add more information regarding the negative impact of COVID-19 pandemic over the medical activity. See and add the following articles: https://doi.org/10.3390/jcm11092452 , https://doi.org/10.3389/fsurg.2022.883935

Thank you for your comments, we have added this reference and now you can read “The World Health Organization (WHO) declared a global pandemic on March 2020, in response to the sudden outbreak the COVID-19, which spread quickly from China across the world [1] and had a very negative impact on the medical activity [2].”

2     Emil-Marian, A.; Reka, K.; Vasile Adrian, M.; Septimiu, V.; Eliza-Mihaela, A.; Eliza, R. Impact of COVID-19 pandemic on Vascular Surgery Unit activity in Central Romania. Front. Surg. 2022, 9, 883935

  1. Please correct the following:

Line 21: impact on HRQOL and on the physical activity level -> impact on HRQOL and the physical activity level

Thank you for your comment. I have corrected the text according to your suggestion and now you can read “The COVID-19 pandemic had a very negative impact on HRQoL and physical activity level of subjects undertaking hemodialysis.”

Line 231: “These authors recommend the promotion of digital-based PA”. Please write the authors with “XY et al. recommend”

Thank you for your comment. As far as I know, it is more correct not to personalize references. Even though, following your suggestion now you can read “Stockwell et al recommend the promotion of digital-based PA [32].”

… etc. Corrections need to be done all over the manuscript.

  1. In the Discussion:

Lines 184-187: Avoid being repetitive. (To our knowledge, this is the first study that shows ……. . To our knowledge, this  is the only study … )

Thank you for your wise comment. We have corrected the sentences to avoid repetitions and now you can read “To our knowledge, this is the first study that shows how negatively the pandemic has impacted this cohort regarding the different subscales of HRQoL measured through the SF-36 and to which extent is decreased the physical activity level. In addition, this is the only study that compares scores of these variables before and after the pandemic.”

  1. Please add a Conclusion section, after the Discussions.

Thank you for your comment, we have added one section, 5. Conclusion

  1. Revise the „Data Availability Statement”.

Thank you for your comment, I will do so

  1. For References:

The references need to be in accordance with the instructions in the template. The list should be with 1., 2., 3., 4. … etc. Also, delete the “[doi]” from the reference list.

Please see the Authors’ Instructions: “In the text, reference numbers should be placed in square brackets [ ], and placed before the punctuation; for example [1], [1–3] or [1,3]. For embedded citations in the text with pagination, use both parentheses and brackets to indicate the reference number and page numbers; for example [5] (p. 10). or [6] (pp. 101–105). The reference list should include the full title, as recommended by the ACS style guide. Style files for Endnote and Zotero are available.”

Thank you for your comment, I have removed doi and brackets from the list

Round 2

Reviewer 1 Report

No more comments.

Author Response

Thank you for your time. Kind regards.

Reviewer 2 Report

Thank you for addressing my concerns. I have a few minor edits to suggest, but otherwise, I am satisfied with the revisions.

The authors responded, “Thank you for your comment. The methodology of the article includes the following information “The present study includes patients from the REalidad Virtual IntraDyalisis (REVID) study. REVID is a currently ongoing randomized trial that investigates the impact of intradialysis exercise. Participants in REVID are enrolled from September 2019 onwards.” When the pandemic started we were in the middle of recruiting participants (data from November to February 2020). Once it started, the HD units were overwhelmed, so we did have no intention to register answers to questionnaires, some of the participants even died. When the situation at the HD units was a bit more stabilized, the nurses there started to register again the questionnaires (November 2020 to February 2021). We add some information to give rationale to the timing of the questionnaires, and now you can read “Thus, the data were recorded before the pandemic, and no more data were recorded until November 2020 once the situation at the dialysis units was under control.” ”

The mentioned rationale for the timing of the questionnaires is an important background of the study population. It is not surprising that the HD units were overwhelmed amid the COVID outbreak, just like other healthcare institutions. Patients in the HD units had more healthcare needs. Therefore, they may have been more susceptible to the negative impact of COVID.

A recent study showed that from June to October 2020, older adults with poor self-rated health experienced a significantly higher delay in seeing specialists than those with fair or good self-rated health. Self-rated health can be considered a simple indicator of HRQOL.

Li, W., Frydman, J. L., Li, Y., & Liu, B. (2022). Characterizing delayed care among US older adults by self-rated health during the COVID-19 pandemic. Preventive medicine, 164, 107308. Advance online publication. https://doi.org/10.1016/j.ypmed.2022.107308

So the authors can say that although it was not possible to administer questionnaires among the recruited HD patients between March 2020 to November 2020, other recent findings suggest that these patients may be negatively affected by the pandemic due to largely possible unmet care needs. So the results should be coherent without the missing time period. I think adding the reference and the discussion can complete the picture of “the fact of living in a pandemic time” among this cohort, which the authors clarified in the response letter as the purpose of this study.

Author Response

Thank you for your comment. We have followed your advice, we have added reference 36 and now you can read at the end of the limitations paragraph

Although it was not possible to administer questionnaires among the recruited HD patients between March 2020 to November 2020, other recent findings suggest that these patients may be negatively affected by the pandemic due to largely possible unmet care needs [36]. So, the results should be coherent without the missing time period

36     Li, W.; Frydman, J.L.; Li, Y.; Liu, B. Characterizing delayed care among US older adults by self-rated health during the            COVID-19 pandemic. Prev. Med. 2022, 164, 107308